# Optical Strain Measurement and Microfractography of the Fractures of Armstal 550 Steel after Temperature Tensile Tests

**DOI:** 10.3390/ma15248875

**Published:** 2022-12-12

**Authors:** Paweł Bogusz, Barbara Nasiłowska, Grzegorz Sławiński

**Affiliations:** 1Faculty of Mechanical Engineering, Military University of Technology, Gen. S. Kaliskiego 2, 00-908 Warsaw, Poland; 2Institute of Optoelectronics, Military University of Technology, Gen. S. Kaliskiego 2, 00-908 Warsaw, Poland

**Keywords:** material strength, microfractography, static tensile test, armour steel, nanoparticles, optical measurements, metallography

## Abstract

A material strength investigation along with a detailed microfractography analysis of fractures formed during static tensile tests of steel Armstal 550 was performed. The tests in this research were conducted in a temperature range of 298 to 973 K. In addition, during tensile tests at ambient temperature, optical measurements of strain maps and the curvature of the neck were performed. The minimum cross-sectional diameter and the radius of the neck curvature during tensile tests were obtained. The data can be directly used to obtain the true stress–strain curve. The material property analysis confirmed the high strength of the Armstal 550 alloy. The ultimate strength at room temperature equals 2.14 GPa, whereas the yield point equals 1.65 GPa. A decrease in the strength parameters along with an increase in temperature was noted. This is a typical phenomenon related to a change in the density and thermal expansion of steel under the influence of the temperature increase. For example, at a temperature of 500 °C, the ultimate strength is more than 50% less than at room temperature. An in-depth analysis of the metallography and microfractography of fractures resulting from static tensile tests showed the formation of atypical nano- and microstructures with an elongated shape. Local nano- and microstructures were observed at different levels of intensity for different temperatures. The largest clusters of nanoparticles were present on the surfaces of the specimens examined at a temperature of 973 K. Scanning microscopy analysis confirmed the presence of molybdenum oxides.

## 1. Introduction

The material properties of high-strength steels are important in the design of modern structures. In particular, this is the case when it comes to armour steel used in military applications. The main advantage of armour steel is its high hardness and strength, combined with a relatively low production cost, which makes it a dominant material for protection against small-arms threats in both military and civil applications. An increase in impact strength is particularly desirable while maintaining the steel’s plasticity [1,2].

The authors did not find any literature references regarding the Armstal 550 alloy’s strength properties. However, in [1,3], a comparison of the strength properties of selected high-performance armour steel alloys can be found. In the case of modern materials, the yield point can exceed 1500 MPa. The exemplary ultimate strength of HCM580 MILAR is equal to 2 GPa. Table 1 also shows Armstal 550 data. The alloy has a yield point of 1600 MPa and an ultimate strength of 1900 GPa, which are considerably high values compared with its competitors. In turn, Kostin et al. [4] investigated Quardian-500 and Armstal 500 high-strength steels. Armstal 500 is similar to the investigated Armstal 550 alloy. The influence of temperature on the structure and durability of the investigated materials was simulated.

Given the lack of strength performance data for Armstal 550 in the literature, unique extensive tensile investigations are presented in the present paper. The presented research can be utilised in a Finite Element Analysis (FEM) with modern advanced constitutive equations, e.g., using the Johnson–Cook model or others. However, these models require strength curves for various temperatures, preferably using the coordinates of true-stress–true-strain to function ideally.

Therefore, additional investigations extended the scope of the research, presented in the form of a diagram in Figure 1. Tensile tests were conducted in a temperature range of 298 to 973 K. During tensile tests at ambient temperature, an optical digital image correlation system (DIC method) was used to measure material deformation. Maps of logarithmic strains were obtained. Logarithmic strains are more relevant in the case of high strains occurring in the necking area of the ductile metal alloy specimen than in the case of engineering strains [5,6].

DIC measurement is currently the most important noncontact strain measurement method and is constantly being further developed [7,8,9,10]. The most significant 3D version is applied to research of standard metals, plastics, thermoplastics [11], advanced polymeric composites [12], biological materials such as tissues [13] and carotid arteries [14] or balloons [15]. It can be observed, therefore, that DIC has become an important method in the field of experimental mechanics [7].

Numerous examples of studies in which material strength investigations and microfractography research were combined to extend the investigations can be found in the contemporary literature [16,17,18,19,20], some of which deal with investigations at different temperature levels.

The microfractography method is one of the key technologies used to clarify the failure mechanisms of machines and structures. The unique properties of nanoparticles are determined by their grain size, shape, structure and intermolecular interactions. Usually, nanoparticles are produced in a controlled environment [21,22,23].

Publications [23,24,25] present the results of tests on cold-drawn pearlitic steel nanowires. The studies conducted by Zhang et al. [23] revealed a phenomenon in which the mechanical properties of nanostructures changed compared to the material at the macroscale. During the process of drawing, the thickness of ferrite and cementite decreases to 20 and 2 nm, respectively. High-angle boundaries parallel to cementite are also formed. Zhang et al. [23] noted that the change in the structure through boundary strengthening and dislocation strengthening is an important mechanism that can influence the properties. Qin et al. [26] presented the production of pearlitic nanostructures through plastic deformation (with the electropulsation method). On the basis of numerical calculations and experimental research, they observed that nanostructured pearlite steel wires have nanoscale lamellae. However, with strong plastic deformations, the pearlitic structure is levelled. This is due to the mobility of carbon, which changes under the influence of mechanical, thermal and electrical states.

Nematollahi et al. [27], in their publication, described the accumulation of carbon (C) atoms in the ferrite layers of pearlitic wires. The process of segregating the components of the crystallographic structure (ferrite, cementite and C atoms) of pearlitic wire nanoparticles made by cold drawing was described in [25,28,29,30,31]. On the other hand, the growth of nanostructures in the form of tin (Sn) whiskers on polished and unpolished copper (Cu) layers after 7 and 60 days was presented by Illés et al. [32].

In [33,34,35], the authors studied molybdenum oxide whiskers produced in the process of synthesis. Merchan-Merchan et al. [34] carried out the synthesis using molybdenum probes placed in an oxy-methane flame. It was observed that not only high temperature but also the oxidising environment influences the growth of molybdenum oxide whiskers [34,35].

However, it is also interesting how nanostructures are made without intentional interference. For example, the separation of nonmetallic inclusions or alloying additives from other components of the structure during the process of generating a separating fracture occurs. Fracture microfractography studies involving static tensile tests are usually described in a classical manner [36,37]. The exceptions are steels [38] or composites [39,40] with characteristic additives that may change the mechanical properties. The results of the research presented in the work of Mazur et al. [38] illustrate the analysis of a microfraction fracture.

Research on the difference in the microfractography of cracks formed depending on temperature changes is lacking in the literature. Therefore, in addition to the main aim of this research, namely, to supplement the existing state of knowledge of the mechanical properties of Armstal 550 steel in a wide temperature range, extensive metallography and microfractography analyses were performed. The research revealed the formation of nanostructures on the fracture surfaces of specimens, which are not described in the literature and have not been subjected to any typical tests.

## 2. Materials and Methods

### 2.1. Subject of the Research

The tests were carried out on Armstal 550 steel. The material was produced and delivered by Huta Stali Jakościowych in Poland [41] in the form of 10 mm thick plates. It was investigated in the as-deposited state. The properties of the alloy are presented in Table 1 and based on [1,3].

Static tensile tests in various temperature ranges, such as 298, 573, 773 and 973 K, were carried out on circular cross-section specimens made of Armstal 550 steel (Figure 2), in accordance with EN ISO 6892-1:2009 [42] and EN ISO 6892-2:2009 standards [43]. The tested elements were made of 10 mm thick sheet metal, from which square bars were cut. Next, the bars were processed with a lathe. The specimens at the ends were threaded, which prevented them from slipping in the holders of the testing machine during the tests.

In the literature [44,45] regarding the true-strain distribution, circular cross-section specimens are the most commonly analysed, as they are in better accordance with theoretical assumptions. In some cases, rectangular sheet specimens are considered, and stress distribution equations are proposed. The authors’ own investigation showed that the results for circular specimens are more accurate [6].

### 2.2. Static Tensile Tests and DIC Measurement Setup

Static tensile tests on Armstal 550 steel were conducted in a wide temperature range: 298, 573, 773 and 973 K (that is, 25, 300, 500 and 700 °C, respectively). Figure 3 presents the testing bed. The investigation was carried out using the Instron 8862 hydraulic universal testing machine (Illinois Tool Works Inc., Norwood, MA, USA) (1) and the Epsilon 3448 linear extensometer (Illinois Tool Works Inc., Norwood, MA, USA) (4). At least three trials were conducted for each temperature level. In the case of ambient temperature, three additional tests were performed with the DIC method. The tests were carried out by controlling the displacement of the machine actuator with the loading rate set to 4 mm/min. The strain rate was about 0.7 × 10^−3^ [1/s]. An Instron CP1117/2 furnace (Illinois Tool Works Inc., Norwood, MA, USA) (2) was used for testing at various temperature levels. Two N-type thermocouples (in both the upper and lower parts) were installed on the specimens and connected to the Instron Eurotherm 2704 controller (3) (Illinois Tool Works Inc., Norwood, MA, USA). The temperature gradient was below 5 °C.

For a precise linear strain measurement, the Epsilon (model 3448) extensometer (Figure 3(4) was used. The equipment is designed for high-temperature measurements of up to 1373 K. The nominal base (initial measurement length) was equal to 0.5 inches (12.7 mm) ± 0.1 inches.

Using a GOM Aramis optical system (Carl Zeiss GOM Metrology GmbH, Braunschweig, Germany) (Figure 4), strain maps were measured on the surfaces of specimens marked with a stochastic black and white pattern. The system is designed for deformation measurements on a material surface during loading. The equipment uses two high-resolution CCD cameras of 2358 × 1728 pixels and a digital image correlation (DIC) method to obtain a three-dimensional image sequence.

A set of cameras equipped with 50 mm lenses were set on a stable tripod and calibrated with a CQ/CP20 55 × 44 mm calibrating plate. The distance between the cameras was set at 88 mm, and the angle between the cameras was 25°. The distance from a loaded specimen was 260 mm. The calibration deviation was 0.029 pixels. In addition, 15 × 15 pixel-sized (about 0.3 × 0.3 mm) facets spaced 13 pixels apart (2-pixel overlay) were used in the DIC calculations. The snapshot frequency was set to 5 Hz. Photos at stage 0 were taken for an unloaded specimen.

A methodology for determining the true stress in respect of the true strain in a standard elastoplastic steel specimen subjected to static axial tensile tests is presented in [5,6]. Tensile specimens made of S235J steel, which was chosen due to its relatively high plastic deformation, were investigated. Strain measurements were performed using traditional extensometers and, additionally, a noncontact optical deformation measuring system. Eventually, displacement fields in the axial and radial directions were determined with the digital image correlation method (DIC).

A similar method was adopted in the Armstal 550 research presented herein. Basic characteristic dimensions describing the curvature of the neck, the width of the neck and the radius of the neck edge curvature were determined. These dimensions are the most common parameters for the calculation of the so-called normalised true stress [5,6]. During tensile tests in the neck, a three-dimensional state of stress was found. Changes in the characteristic dimensions of the neck can be determined by applying Bridgman’s and other researchers’ formulas for stress distribution in the neck. Moreover, a strength analysis extended to include a temperature test can be utilised in Johnson–Cook Finite Element Modelling simulations.

### 2.3. Characterisation by Scanning Electron Microscopy

The microfractography surface was investigated using Scanning Electron Microscopy (SEM) using a Quanta 250 FEG SEM (FEI, Hillsboro, OR, USA). SEM images were acquired using a backscattered detector (ETD-BSE, FEI, Hillsboro, OR, USA) with an accelerating voltage of 10 kV.

## 3. Results and Discussion

### 3.1. Metallographic Examination of the Material

A metallographic examination of the surface of Armstal 550 steel in the as-deposited state was performed using the Quanta 250 FEG microscope and showed the presence of a martensitic structure (Figure 5), which is typical for armour steel alloys [1]. The study of the elemental composition showed the following chemical composition (in %): Mn 2.30; Si 0.64; P 0.01; S 0.0020; Cr 1.65; Ni 2.88; Cu 0.02; Mo 1.07; V 0.09; Al 0.16; Ti 0.76; and Fe = bal.

On the surface of the metallographic sample, apart from the basic alloying elements, i.e., iron (Figure 6b,c), non-metallic inclusions of molybdenum and titanium (Figure 6d,e) were observed.

### 3.2. Static Tensile Tests Results

During the static breaking test of Armstal 550 steel in a temperature range of 298, 573, 773 and 973 K, the stress–strain curves and fundamental material properties (yield strength Re, tensile strength Rm, destructive deformation Ru and modulus of elasticity E) were determined. The results of the tests, together with the statistical analysis, are presented in Table 2. The analysis of the results of the static tensile test showed that the scatter of the results was below 1% of the statistical error.

The high-performance material properties of Armstal 550, as summarised in Table 1, were confirmed. The tensile strength is 2.14 GPa, and the yield strength is over 1.6 GPa. The modulus of elasticity is typical for steel.

As a result of the temperature increase during the static tests, a decrease in mechanical properties was observed. The strength and yield point at temperatures of 573, 773 and 973 K compared to the ambient temperature data decreased by about 30, 50 and 90%, respectively. In turn, the modulus of elasticity decreased by 16, 22 and 74% compared to the value at ambient temperature. A graph illustrating the dependence of mechanical properties on the temperature is presented in Figure 7. The experimental points were approximated by second-degree polynomials, as the R-squared parameter was close to one. A Poisson ratio equal to 0.284 was obtained during additional strain gauge measurements.

The σ-ε tensile curves of the exemplary specimens are shown in Figure 8. It was observed that with a temperature increase, the tensile curve of Armstal 550 steel decreased. Armstal 550 armour steel does not show a clear yield point. A yield strength of R0.2% was determined. For a temperature of 973 K, there was a significant drop in strength. However, the fracture strains increased significantly.

Logarithmic axial strain (so-called true strain) maps in the area of the neck were obtained as the DIC measurements were taken. Logarithmic strains are more relevant in the case of high strains that occur in the necking area of the specimen than engineering strains.

The DIC method allowed the acquisition of a sequence of stages that detail the course of the axial tensile process in the formation of a neck, from the beginning of the test to the fracture (Figure 9). The first photo shows an unloaded specimen (a). Figure 9g shows the specimen prior to separation, and Figure 9h shows it after the crack formed. Stage numbers are given. A map of axial logarithmic strains, as well as the position and length of the virtual extensometer (yellow axial lines), is presented against the background of each of the photos. The extensometer length shown in the first frame represents the starting value for that extensometer (25.1 mm). It is shown that the logarithmic strain in the neck just before the fracture reaches a value of about 41.6%.

The next step was the determination of the basic characteristic dimensions describing the curvature of the neck, that is, the width diameter of the neck and the radius of the neck edge curvature. These dimensions are the most common parameters for the calculation of the so-called normalised true stress, which takes into account the fact that during the tensile test in the neck, a three-dimensional state of stress occurs. The data can be directly used to obtain the true stress–strain curve, which is especially useful for FEM modelling.

The diagrams in Figure 10 show the course of the neck dimensions as a function of the axial engineering strain. The data were obtained according to the procedure described in [6]. The initial specimen diameter of 4.94 mm shrank to about 3.77 mm (Figure 10a). This dependency is approximately bilinear. The axial stress related to the actual minimal neck cross-section area reached 2.77 GPa at a maximum.

The radius of the neck curvature was determined on the basis of circles built in the area where the neck was formed [6]. The neck curvature started to form from an axial strain of about 0.04, and then it was seen to be equal to 450 mm (Figure 10b). This is the phase when the yielding point was clearly exceeded and the ∂σ/∂ε ratio equalled zero. Before this point, the sample has a uniform cross-section, and the radius of the curvature tends to infinity. At the end of the test, in the advanced necking phase, and prior to the break, the average radius of the neck was equal to about 45 mm.

Additionally, a hardness test was carried out. The hardness tests were performed on a universal hardness tester (INNOVATEST Nexus 7000, INNOVATEST Europe BV, Maastricht, Netherlands). Three measurements of Rockwell hardness were carried out at various places on the sample plate. The average Rockwell hardness was 52.5 HRC, which gives a Brinell hardness of about 527HB. The material hardness measured after the preparation of the specimens (milling and grinding) was lower than the hardness data presented in Table 1.

### 3.3. Microfractography of Fractures after Static Tensile Tests

Figure 11 presents the specimens of Armstal 550 steel after static tensile tests were conducted at different temperature levels: 298 (Figure 11a), 573 (Figure 11b), 773 (Figure 11c) and 973 K (Figure 11d). The increase in temperature from 773 to 973 K caused the strong oxidation of the surface. This was due to the lack of a protective gas atmosphere, which, according to standards [42,43], was not required for the research conducted. Therefore, the specimens tested at temperatures of 773 (Figure 11c) and 973 K (Figure 11d) showed surface slagging.

In Armstal 550 steel, the carbon content exceeds 0.25% (0.38%) (Section 2.1), which strongly influences the formation of hardened structures. According to Kostin et al. [4], in the Armstal steel alloy, with increasing temperature, the α region is extended by the presence of chromium (0.52%) and vanadium (0.004%), while nickel (1.91%) lengthens the γ region. As a result, changes in the position of Ac3 and Ac4 temperatures arise in the thermogenetic diagrams of the transformation of supercooled austenite. However, elements that do not form carbides, for example, aluminium (0.029%), nickel, silicon (1.91% + 0.21%) and copper (0.09%), do not cause significant changes in the shape of the S curve (of isothermal transformations of austenite) either. In fact, they only shift the transition temperature and increase the durability of austenite by shifting the lines in the diagram to the right. At the same time, titanium (0.037%), chromium, vanadium and molybdenum (0.52 + 0.004 + 0.58%) influence the kinetics of the isothermal transformation. At different temperatures, the influence of these elements on the rate of austenite decomposition is variable and leads to the clear separation of the pearlite and bainite transformations [4,46]. Thus, the contents of carbon and the main alloying elements in Armstal 550 cause changes in the structure and mechanical properties in various temperature ranges.

Microphotography of the fracture surface after the static tensile test was performed on the surfaces of separated fragments of Armstal 550 at temperatures of 298 (Figure 12a), 573 (Figure 12b), 773 (Figure 12c) and 973 K (Figure 12d). As the temperature increased, the diameter of the separating fracture decreased due to the high plasticity of the material at higher temperatures. The crack in Armstal 550 expanded as a result of a static ductile failure by several millimetres before the final failure. The crack preceded the formation of the neck. As the temperature increased, the diameter of the neck decreased. On the surface of the fractured materials tested at a temperature of 298 K, parallel lines with numerous inclusions at the bottom were observed. These were caused by the rolling process used in the production of the Armstal 550 steel sheet.

Figure 13 presents fracture fragments tested at different temperatures: 298 (a), 573 (b), 773 (c) and 973 K (d). In all cases, both slip and breaking fractures occur in the core. In the samples tested at temperatures of 298–773 K (Figure 13a–c), a clear boundary between the separation crack and the slipping area was observed, which is also visible in Figure 12a–c. However, at a temperature of 973 K, no separation of the two zones was observed (Figure 12d and Figure 13d). This is related to an increase in the ductility of Armstal 550 at high temperatures.

A microfractography analysis was performed on the entire surface. However, due to the need for systematisation, the results of the research (Figure 14) are focused on the central part of the specimen. The observation of the fracture microfractography showed the presence of a network of ridges and cavities with a cup-conical character at temperatures of 298–573 K. The appearance of a malleable failure has been described as microvoid coalescence by Moore et al. [37].

During the microfractography analysis (Figure 14), it was observed that the coalescence of microspheres occurred around fine inclusions (Figure 14a–d). The voids that formed, which increased during the plasticisation of Armstal 550 steel, were the largest at a room temperature of 298 K (Figure 14a,b). They had the character of a honeycomb structure, as described by Katarzyński et al. [36].

As the temperature increased (573 K), small spaces joined together, creating, apart from the fine structures of microvessels (Figure 14d), larger local voids (shown by arrows in Figure 14c). The microstructure ridges between the voids were the largest at room temperature.

On the fracture surface after the static tensile test at temperatures of 773–973 K (Figure 14e–h and Figure 15a–c), the presence of nanostructures in the form of whiskers was observed. Scanning microscopy analysis with an etched EDAX (Energy-Dispersive Spectroscopy) detector showed that they were molybdenum oxides (Figure 6). The greatest number of these nanoparticles was visible in fractures formed at a temperature of 973 K (Figure 14h and Figure 15b,c). This research showed that at a temperature of 773 K, the molybdenum oxide nanoparticles were 357–719 nm long and 15.79–263 nm wide, while at 973 K, their dimensions were 3.231–729 nm in length and 24–71 nm in width.

The nanostructure formed by elongated nanoparticles (whiskers) of molybdenum oxide, which originated during the static tensile test, is an interesting phenomenon, hitherto rarely described in scientific publications. A scanning microscopy analysis with the etched EDAX detector showed that they were molybdenum oxides (Figure 6). Typically, during the formation of a separating fracture, the surface of which depends on many factors, including the load, sample shape and material structure, a brittle or ductile fracture arises. However, in the case of Armstal 550 steel, a nanostructure was observed, which is not described in the literature, nor has it been subjected to any typical tests [36].

## 4. Conclusions

Extensive research into the mechanical properties of Armstal 550 at various temperatures and a microfractography study of the surfaces of fractures formed after static tensile tests were carried out. The following conclusions were drawn:Armstal 550 is a high-performance armour steel alloy. At ambient temperature, the ultimate strength is equal to 2.14 GPa, the yielding point is equal to 1.65 GPa, and the Poisson ratio is 0.284. The average elastic modulus is 205.2 MPa.On the fracture surface of the static tensile test at temperatures of 773–973 K, the presence of a local nanostructure in the form of whiskers was observed, which have hitherto been rarely described in scientific publications. Scanning microscopy analysis with the etched EDAX detector showed that they were molybdenum oxides.No growth of molybdenum oxygen nanoparticles was observed at 298–573 K.High agreement with the DIC measurement data was obtained for three specimens investigated. The maximum axial stress related to the actual neck cross-section area, the so-called true axial stress, is 2.77 GPa.In an advanced necking phase, the radius of the neck curvature is equal to about 45 mm. The initial specimen diameter of 4.94 mm shrank to about 3.77 mm.

## Figures and Tables

**Figure 1 materials-15-08875-f001:**
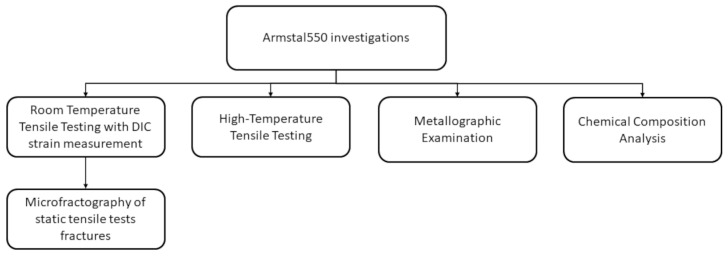
A schematic diagram presenting the scope of the research.

**Figure 2 materials-15-08875-f002:**
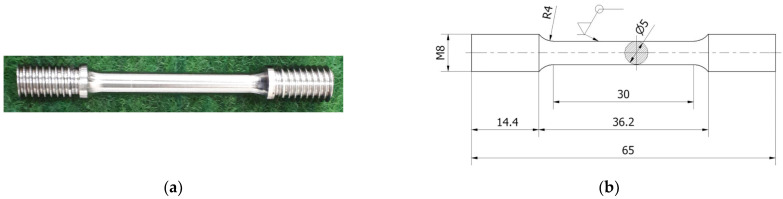
Specimen of Armstal 550 steel: (**a**) a photo of a specimen; (**b**) a technical drawing.

**Figure 3 materials-15-08875-f003:**
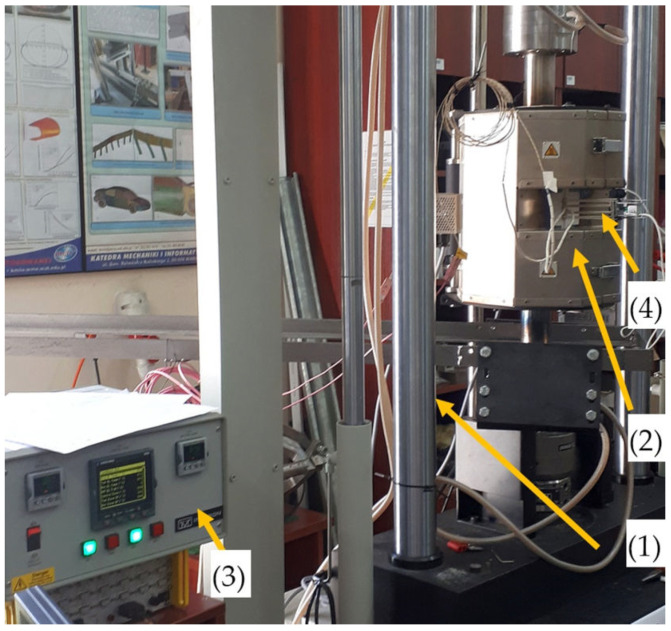
Testing stand: (1) Instron 8862 testing machine, (2) high-temperature furnace, (3) Eurotherm 2704 controller, and (4) Epsilon 3448 linear extensometer.

**Figure 4 materials-15-08875-f004:**
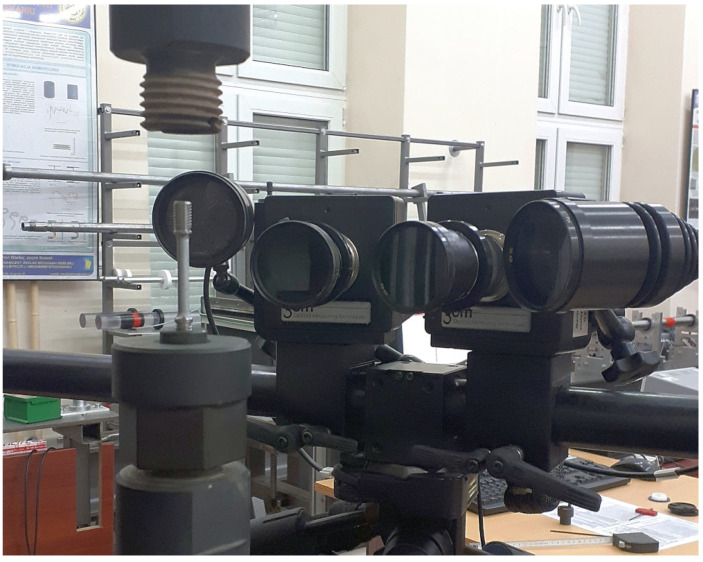
Installation of an exemplary specimen in a threaded holder of the Instron 8862 machine along with the setup of the 3D cameras of the DIC measurement system.

**Figure 5 materials-15-08875-f005:**
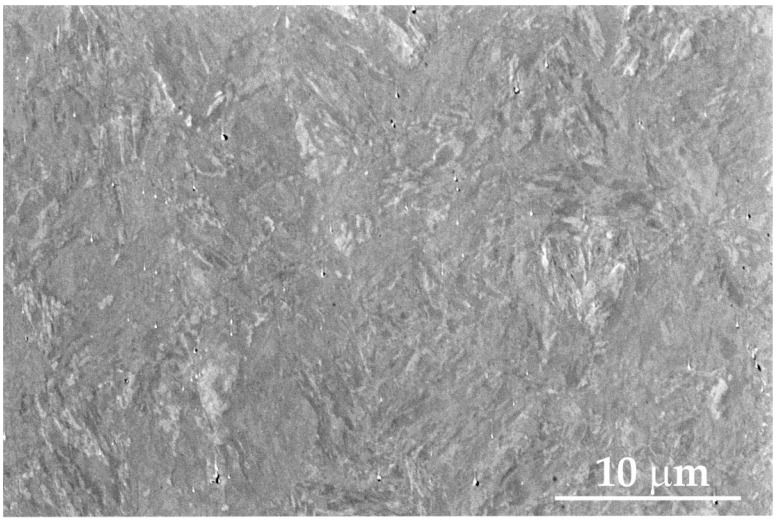
SEM picture of the surface of Armstal 550 steel with martensitic structure.

**Figure 6 materials-15-08875-f006:**
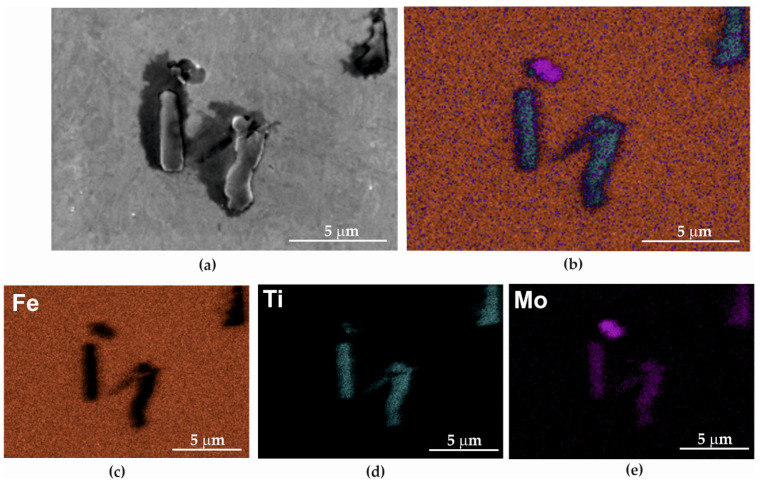
Metallographic examination of Armstal 550 steel: (**a**) examined surface; (**b**,**c**) iron; (**d**) inclusions of titanium; and (**e**) molybdenum.

**Figure 7 materials-15-08875-f007:**
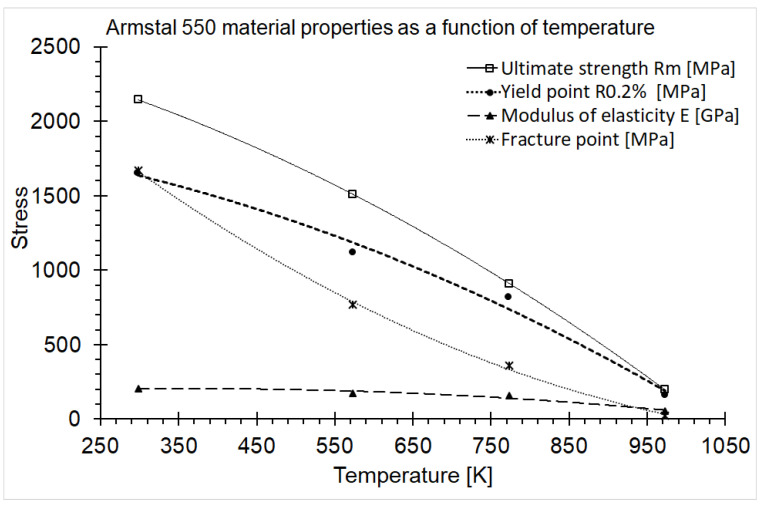
Graph of dependence of Armstal 550 mechanical properties on temperature.

**Figure 8 materials-15-08875-f008:**
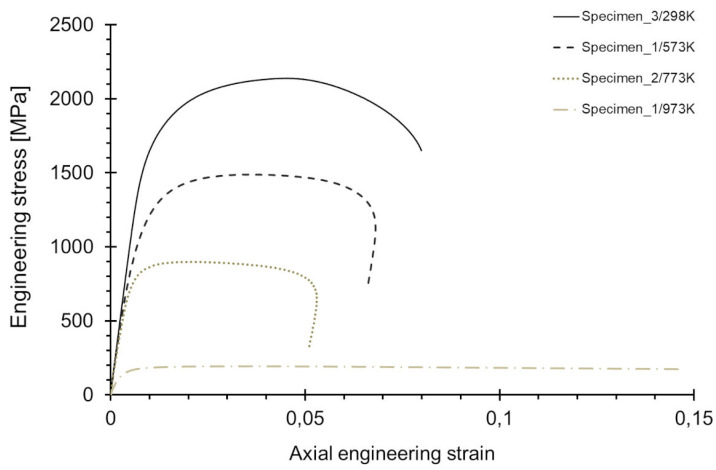
Comparison of the σ-ε tensile curves for the temperature levels tested.

**Figure 9 materials-15-08875-f009:**
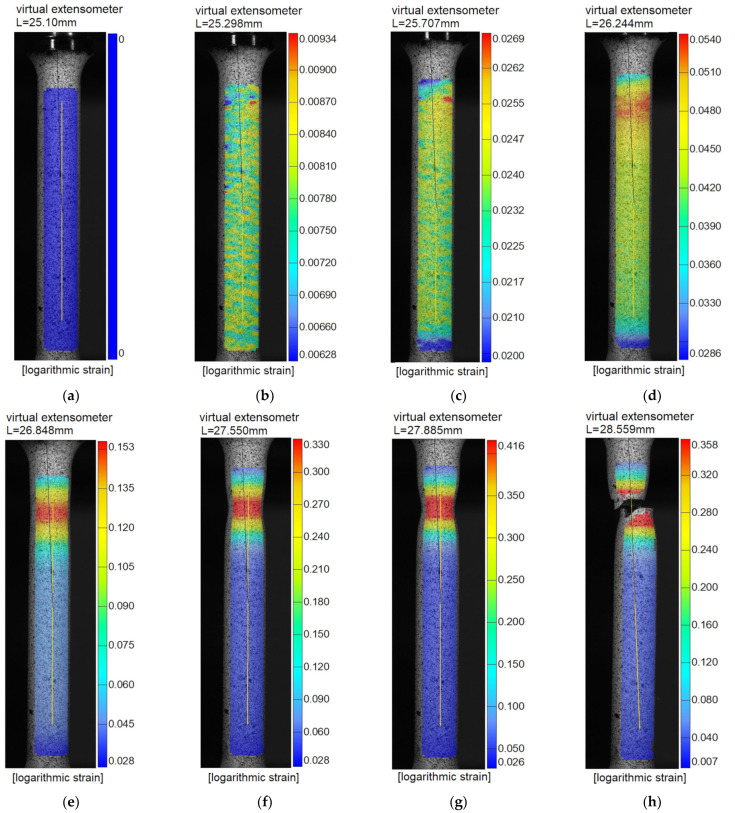
Stage photos recorded with DIC method: (**a**) stage 0—unloaded specimen; (**b**) stage 50; (**c**) stage 100; (**d**) stage 150; (**e**) stage 200; (**f**) stage 250; (**g**) stage 272—prior to separation; (**h**) stage 273 (after fracture).

**Figure 10 materials-15-08875-f010:**
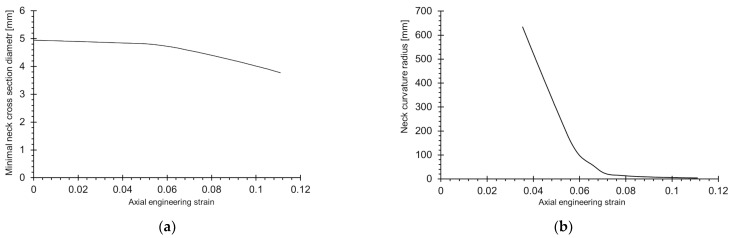
Changes in characteristic dimensions of neck during tensile test: (**a**) minimum neck cross-sectional diameter and (**b**) neck curvature radius.

**Figure 11 materials-15-08875-f011:**
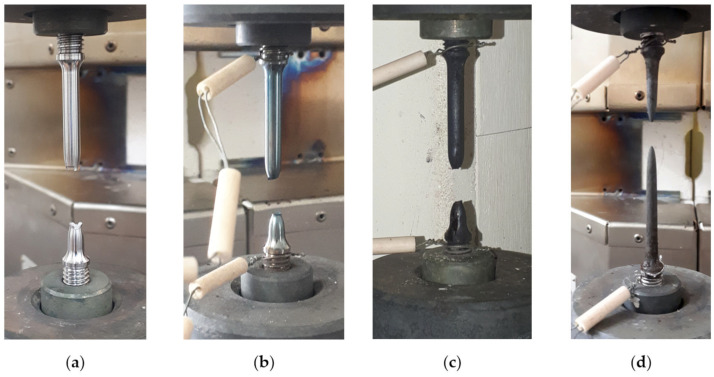
Tensile fractures of Armstal 550 steel specimens after tests conducted at high temperatures: (**a**) 298 K; (**b**) 573 K; (**c**) 773 K; (**d**) 973 K.

**Figure 12 materials-15-08875-f012:**
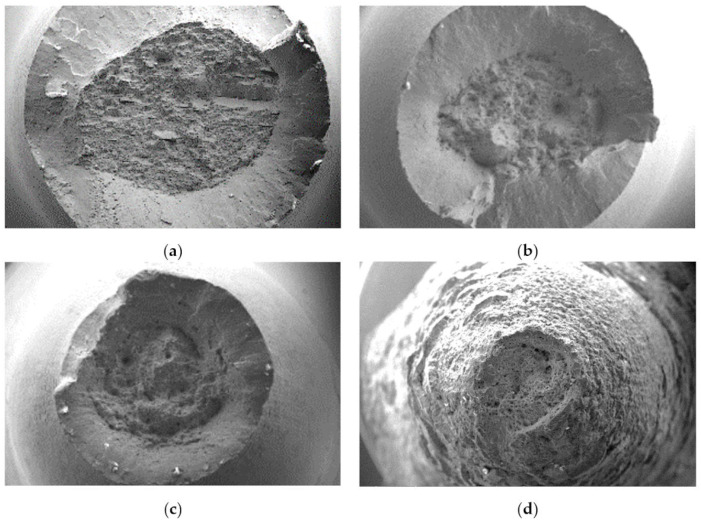
Microfractography of the tensile separating fractures of Armstal 550 specimens tested at temperatures of (**a**) 298 K, (**b**) 573 K, (**c**) 773 K, (**d**) 973 K.

**Figure 13 materials-15-08875-f013:**
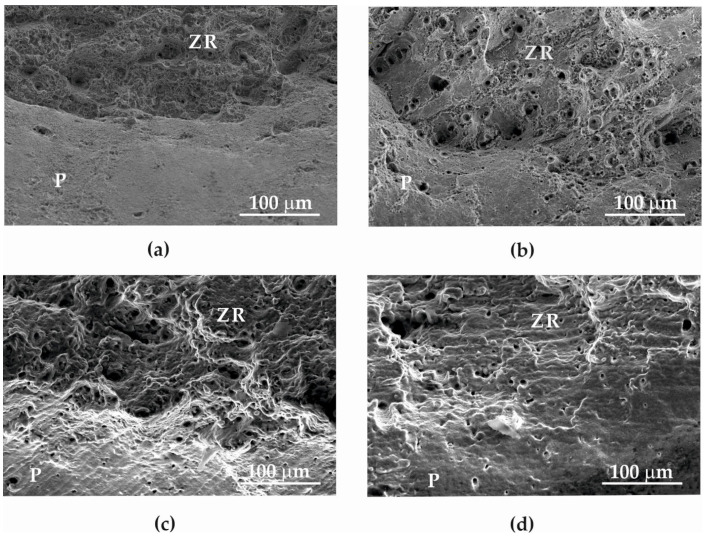
Microfractography of Armstal 550 separating fracture surfaces, in which both slip (P) and separating cracks of the core (ZR) occur after static tensile temperature tests at (**a**) 298 K, (**b**) 573 K, (**c**) 773 K, (**d**) 973 K.

**Figure 14 materials-15-08875-f014:**
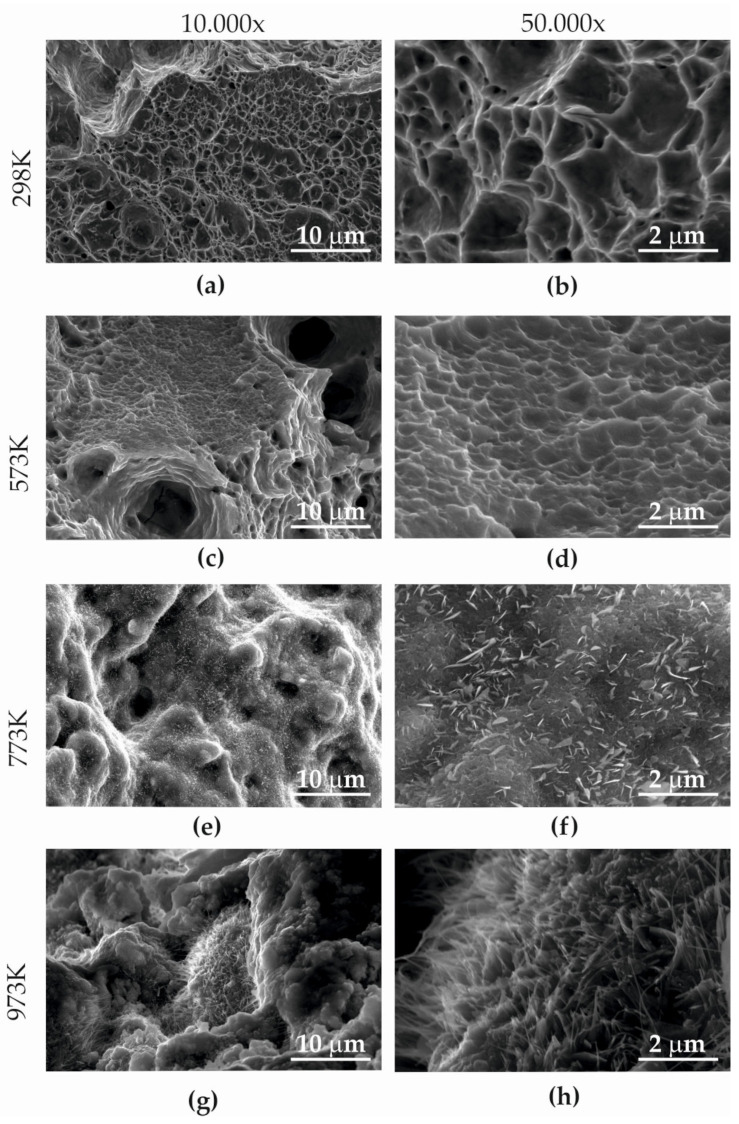
Microfractography of Armstal 550 separating fracture surfaces after static tensile tests at various temperatures: (**a**,**b**) 298 K; (**c**,**d**) 573 K; (**e**,**f**) 773 K; (**g**,**h**) 973 K.

**Figure 15 materials-15-08875-f015:**
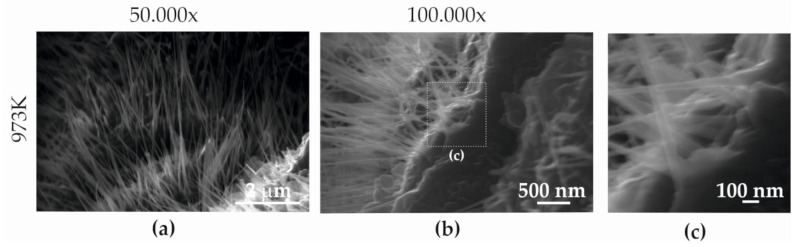
Thread surface microfractography of Armstal 550 steel specimens after tests conducted at temperatures of (**a**) 773 K and (**b**,**c**) 973 K.

**Table 1 materials-15-08875-t001:** Comparison of properties of selected high-performance armour steel alloys based on [1,3].

Steel Description	Plate Thickness	Yield Strength R0.2%	Ultimate Strength	Hardness HB
Armox 500T	6.13	1300	1500–1750	480–540
Armox 560T	to 100	1300	1600–1900	530–580
Armstal 550	to 20	1600	1900	≥550
Hardox 500	to 50	1300	1550	450–530
HCM580 MILAR	8–40	1500	2000	580–620
XH 129	4–25	1400	1700	477–534
2P (GOST-21967 s)	to 25	~1350	~1600	444–514
30 PM	to 20	1300	1600	460–540

**Table 2 materials-15-08875-t002:** The results of tensile tests of Armstal 550 armour steel for all temperature levels.

Description	Temperature [°C/K]	Ultimate Strength Rm [MPa]	Yield Point R0.2% [MPa]	Modulus of Elasticity E [GPa]	Fracture Point Ru [MPa]
Average	25 °C/298 K	2143.7	1653.0	205.2	1668.6
Std dev.		4.5	19.1	2.8	7.5
Std error [%]		0.12	0.67	0.78	0.26
Average	300 °C/573 K	1510.9	1116.7	171.5	766.7
Std dev.		18.7	35.1	2.4	24.6
Std error [%]		0.71	1.82	0.82	1.85
Average	500 °C/773 K	910.4	817.3	159.2	357.5
Std dev.		10.3	10.8	3.8	10.3
Std error [%]		0.65	0.76	1.39	0.65
Average	700 °C/973 K	196.5	162.0	53.0	23.7
Std dev.		1.7	2.0	2.7	0.1
Std error [%]		0.50	0.71	0.29	0.25

## Data Availability

Not applicable. The necessary data are contained within the article.

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
