# Peer review of "Optical Strain Measurement and Microfractography of the Fractures of Armstal 550 Steel after Temperature Tensile Tests"

_materials, 2022, doi:10.3390/ma15248875_

Round 1
Reviewer 1 Report
This paper investigated the mechanical behavior of Armstal 550 steel at a temperature of 298K to 973K by optical strain measurement and microfractography. This paper is more like an experimental report rather than a research paper. Firstly, the Introduction Section is not well organized. Secondly, only the fracture surface of the studied steel was characterized via SEM. The detailed microstructure characterization of the studied steel is missing. Thirdly, the DIC method is a well-developed technique to capture the strain evolution of tensile deformation in extensive researches. I fail to grasp the scientific novelty and impact of the reported results. Last but not least, the EDAX results are not presented to support the nano-whiskers being molybdenum oxides.
Author Response
Dear Reviewer,
Please find the attached filed with Review Responses.
Best regards.

Reviewer 2 Report
Manuscript entitled “Optical strain measurement and microfractography of the fractures of Armstal 550 steel after temperature tensile tests” could be interesting for the readers. However, the paper needs significant revision before publication. I have listed a few comments that need to be addressed:
1. Add some concrete results in the abstract section.
2. In abstract authors claimed “formation of atypical nano and microstructures” add concrete evidence in support of this statement.
3. What is the novelty of this work that should be clearly presented in the introduction?
4. There are no details about the materials used, add them with specifications.
5. Write the full form once when mentioning for the first instance.
6. Add a schematic diagram to show the overall work.
7. Improve the quality of the Figures.
8. The integration of the results from different parameters should be improved carefully.
9. The obtained results were not discussed well at all, it must be improved with previously published literature.
10. Section 4: Discussion and conclusion should be changed to Conclusion.
11. Also, carefully revise the typos and linguistic errors to make the manuscript error-free.
Author Response
Dear Reviewer,
Please find the attached filed with Review Responses.
Best regards.
Authors

Round 2
Reviewer 1 Report
The authors have made revisions accordingly based on the comments. This version satisfies the requirements of the present publication.